# Competitive Advantage of *Broussonetia papyrifera* Growing in a Native Area as Suggested by Structural Diversity

**DOI:** 10.3390/biology12111410

**Published:** 2023-11-09

**Authors:** Yanrong Zhou, Guangfu Zhang

**Affiliations:** Jiangsu Key Laboratory of Biodiversity and Biotechnology, School of Life Sciences, Nanjing Normal University, Wenyuan Road, Nanjing 210023, China; zyr1711693133@163.com

**Keywords:** biological invasion, competition, paper mulberry, regenerating strategy, structural diversity

## Abstract

**Simple Summary:**

In this study, we ask whether the invasive paper mulberry (*Broussonetia papyrifera*) has a competitive advantage over neighbor trees in a native area. We determined the competitive capacity of paper mulberry in three deciduous broad-leaved forests using structural indices including the mixing index, the tree–tree interval index, and the diameter/height differentiation index. It was found that the reference paper mulberry had a slight competitive advantage over neighboring trees in both the horizontal and vertical planes. Such a competitive ability may play a significant role in the ecological invasion of paper mulberry. Our findings not only help to reveal the invasion mechanism of paper mulberry, but also provide an important reference for the management and utilization of paper mulberry in invaded areas.

**Abstract:**

Paper mulberry (*Broussonetia papyrifera*) is currently an invasive species on several continents. However, little is known about whether paper mulberry has a competitive advantage over its surrounding trees in its native distribution range, subtropical regions of China. Here, we determined the relative intraspecific and interspecific competitive capacity of paper mulberry in three subtropical deciduous broad-leaved forests using the indices of structural diversity including the mixing index, the tree–tree interval index, and the diameter/height differentiation index. It was found that more than 80% of mingling index values were not greater than 0.25, suggesting a stronger competitiveness of paper mulberry relative to other tree species. The tree–tree interval index values ranged between 1 m and 2 m, suggesting a strong competition between paper mulberry and its neighbors. Moreover, more than 60% of the height differentiation index and diameter differentiation index values were positive, suggesting that the reference paper mulberry had a slight competitive advantage over neighboring trees in both the horizontal and vertical planes. These collectively suggest a competitive advantage over other tree species in the native distribution range, which may play a significant role in the ecological invasion of paper mulberry. Our findings not only help to reveal the invasion mechanism of paper mulberry, but also provide an important reference for the management and utilization of paper mulberry in invaded areas.

## 1. Introduction 

*Broussonetia papyrifera* (L.) L’Hér. ex Vent., which is known as paper mulberry, is a deciduous tree species from the family Moraceae. This species can grow up to 10–20 m in its native regions. Paper mulberry is perennial and dioecious; its male inflorescences are long spicate, 3–8 cm, and its female inflorescences are globose and orange-red, 1.5–3.0 cm in diameter when mature [1,2]. It has economic, medicinal and environmental values. Its bark fibers can be utilized for the production of paper, cloth, and rope [3], and the wood can be used for furniture; leaves for animal fodder [4]. The leaves, fruit, and bark can be used as medicines [5,6]. In addition, it is used for Mn-contaminated soil treatment because of its high tolerance to Mn stress [7].

Paper mulberry is native to Eastern Asia, and it occurs in many countries like China, Cambodia, Japan, Korea, Laos, Malaysia, Myanmar, Thailand, and Vietnam [1]. For example, this species is widely distributed in 21 provinces of China, including Anhui, Jiangsu, and Fujian [1]. Over the centuries, it has been intentionally introduced to Europe, America, Africa, and Pacific Islands for economic and aesthetic purposes [8,9]. Unfortunately, this species has become invasive on several continents and in more than ten countries including Pakistan [10], India [11], Argentina [12], the United States [13], and Ghana [14]. Such a woody invader poses threats to native tree species, forest vegetation, and timber production, and it even causes pollen-related allergies for local people [9,10]. For example, based on comparative analyses and removal experiments, Bosu et al. [14] concluded that paper mulberry invasion had largely reduced the cover and abundance of indigenous broadleaf species in Ghana’s forest ecosystems.

Why can paper mulberry invade many parts of the world? On the one hand, this tree species is able to adapt to various climatical and edaphic conditions throughout the world [13]. It is therefore possible for such a tree species to invade various habitats within the introduced areas which are often subject to anthropogenetic disturbance. For instance, paper mulberry was introduced as an avenue tree species in Pakistan in 1960s, and it has become a successful invader in many regions of the country since 1990s [15]. On the other hand, the invasion success of paper mulberry is related to its biological characteristics including rapid growth rate, vegetative regeneration strategy, effective propagule dispersal and allelopathy [6,16].

Existing literature indicates that the invasion success of paper mulberry may be mainly accounted for by its genetic diversity, competitive ability and chemical allelopathy. Liao et al. [17] reported that there was a high genetic diversity of paper mulberry at the species level detected by ISSR (inter-simple sequence repeat) molecular marker. High genetic diversity could enhance the adaptability of a species and thereby facilitate its successful invasion outside the original distribution. However, Peñailillo et al. [18] revealed that most of the current paper mulberry plants in the Pacific Islands appeared to be descended from female clones which were introduced prehistorically, resulting in the absence of sexual reproduction [19]. Similarly, Zenni et al. [20] pointed out that genetic diversity may not be the critical factor resulting in successful invasion of non-native populations. 

Indeed, Maan et al. [9] suggested that the displacement of native vegetation by paper mulberry invasion resulted from competitive exclusion or allelopathy in India. Ssemanda and Ssekuubwa [21] found that paper mulberry invasion reduced the cover and basal area of native timber tree species in Africa tropical forests; Bosu et al. [14] noted that the paper mulberry invasion reduced the cover of resident tree species in Ghana forest stands, although the species composition between invaded and uninvaded plots was similar. Nevertheless, their findings cannot be explained by allelopathy, since successful invaders due to allelopathy “often establish virtual monocultures where diverse communities once flourished, a phenomenon unusual in natural communities [22]”. Thus, we argue that the competitive capacity of paper mulberry may be one of the most important factors leading to its invasion in the introduced area.

Studies addressing the competitive capacity of paper mulberry have been exclusively conducted in the invaded areas [14], and little is known about the competition of this tree with its surrounding species in the native region. Studies often suggest that invasive species become competitive only after invading a region [23,24]. However, it is possible that species that are more competitive in its native areas are more likely to be invasive outside its natural distribution.

There are two categories of methods determining plant competition among tree species. One is to culture plants in common garden experiments with a partial additive or replacement design. This method is generally used for seedlings and saplings [25,26]. The other is to survey in the field, which is suitable for large plants, especially for tree species in forests [27,28]. In particular, quadrat-free approaches have been developed and widely applied to determine the competitive capacity of forest trees in recent years (e.g., [29,30]). 

Structural indices have been proposed to facilitate the application of the quadrat-free method. The structural indices, including the mixing index, the tree–tree interval index, and the diameter/height differentiation index, can be used to characterize spatial structure of forests [31,32]. The competition between the reference tree and its neighbors in the horizontal and vertical direction can be analyzed by these four structural indices. Although the quadrat-free approach is low-cost and time-saving, it can accurately show the growth status of the reference tree and its neighbors, reflecting the competition outcome between forest trees [33,34]. For example, by the quadrat-free approach, Liu et al. [30] measured the structural diversity of *Parrotia subaequalis* in the forests of eastern China, and revealed a strong interspecific competition between the endangered tree species and its associated trees.

Here, in three deciduous broad-leaved forests of eastern China, the native distribution region of paper mulberry, we determined the intraspecific and interspecific competitive capacity of paper mulberry using the structural indices. We asked whether paper mulberry was more competitive relative to its surrounding trees in the original area. Specifically, we aimed to: (1) explore the competition of paper mulberry in terms of the mixing index, the tree–tree interval index, and the diameter/height differentiation index, and (2) compare the competitive capacity of paper mulberry females and males using the mixing index and average interval index, respectively. The objective of this study is to explain why paper mulberry is widely distributed in China, and shed light into the underlying mechanisms of the invasion outside its original range.

## 2. Materials and Methods

### 2.1. Study Area

Paper mulberry is a widespread tree species (Figure 1a,b) in subtropical–temperate climates [35], covering more than 20 provinces (Figure 2a). Based on our field survey, we selected three deciduous broad-leaved forests with paper mulberry, which are located in Xianlinhoushan Mountain (for short, XLHS) in the suburban areas of Nanjing, Jiangsu Province, eastern China (Figure 2b). XLHS includes many low mountains and hills (32°6′ N, 118°54′ E), with an average elevation of 45.0 m. According to Köppen’s climate classification [36], this region belongs to the subtropical humid climate zone, with four distinct seasons, a mild climate and adequate rainfall. The mean annual temperature of XLHS is 15.4 °C, with the extreme maximum being 39.7 °C in July and the extreme minimum being −13.1 °C in January, and the mean annual precipitation is 1106 mm [37,38]. The main soil type is of yellow-brown soil.

The paper mulberry forests are usually distributed on sunny slopes of XLHS. These forests can be roughly divided into three layers in vertical structure [37,39]. The tree layer is dominated by *B. papyrifera, Pistacia chinensis* Bunge, *Rhus chinensis* Mill, *Maclura tricuspidata* Carriere and *Celtis sinensis* Pers., which are sometimes associated with *Vitex negundo* var. *cannabifolia* (Sieb. et Zucc.) Hand.-Mazz., *Melia azedarach* L., *Robinia pseudoacacia* L., and *Pinus massoniana* Lamb. *Robinia pseudoacacia* has become a naturalized species in China over the past several decades after introduced from Germany [40,41]. The shrub layer is dominated by *Rubus lambertianus* Ser., *Rosa multiflora* Thunb., *Sageretia thea* (Osbeck) Johnst., *Rubus parvifolius* L., *Rhamnus globosa* Bunge, which are associated with saplings of some tree species, such as *B. papyrifera*, *C. sinensis*, and *R. chinensis*. The herbaceous layer is dominated by *Trachelospermum jasminoides* (Lindl.) Lem., *Arthraxon hispidus* (Thunb.) Makino, *Polygonum hydropiper* L., which are occasionally associated with a few seedlings of trees and shrubs.

### 2.2. Field Sampling

Based on our field survey, we established three paper mulberry forests (namely plot A, plot B and plot C). Each plot was 30 m × 24 m in size and each plot was divided into 45 4 m × 4 m subplots. Following the sampling approach of Ruprecht et al. [42], a structure unit, consisting of a reference tree (or a reference point), and four nearest neighboring trees in the vicinity of the tree (point), was designed to measure the structural diversity of a particular tree cohort [29,43]. Namely, a single paper mulberry tree, which was the closest to the center of a subplot, was identified as a reference tree at each subplot, and four trees closest to the reference tree were identified as neighboring trees. We established a total number of 41, 41, 43 sampling points (i.e., reference trees, or structure units) in plot A, plot B and plot C, respectively. Because no paper mulberry tree was found in some subplots, the number of sampling points was smaller than the number of subplots. The reference trees and their neighboring trees were at least 4.0 cm of diameter at breast height (DBH). When there was a neighboring tree outside the boundary of the subplot, the second nearest neighbor was selected so that five trees were in the structure unit [29,44]. At each subplot, we recorded the tree species, number, diameter, height of all selected trees and the distance between a reference tree and its neighbors (Appendix A). 

### 2.3. Structural Indices

Structural indices were applied to reveal the intraspecific and interspecific competition of paper mulberry populations in the three plots (Table 1).

The mixing index (*M_i_*) describes the interspecific and intraspecific competition of species in a forest [30,32]. The index was calculated as:(1)$$M_{i}=\frac{1}{n}\sum_{j=1}^{n}v_{ij}$$

When the reference tree (*i*) and its one neighbor (*j*) are the same species, the value of *v_ij_* was 0. If they were different, the value was 1. And *n* was the total number of neighboring trees (4). Consequently, *M_i_* had five values (0.00, 0.25, 0.50, 0.75, 1.00). The smaller the value of *M_i_* was, the higher probability that neighbors were as same species as the reference tree [41]. In contrast, the greater the *M_i_* was, the more different tree species were intermingled with paper mulberry. In this case, the competitive capacity of the reference trees was negatively associated with *M_i_*. 

The tree–tree interval index (*D_i_*) describes the average distance between reference tree and its neighbors, reflecting the density of the forest [42]. The index was calculated as:(2)$$D_{i}=\frac{1}{n}\sum_{j=1}^{n}s_{ij}$$
*s_ij_* is the distance between the reference tree (*i*) and its neighboring trees (*j*); *n* is the number of neighboring trees. The larger the *D_i_* was, the farther the distance between these trees. In contrast, the smaller the *D_i_* was, the more intense the competition between them was. In this case, the intensity of the competition between the reference tree and its neighbors was negatively associated with *D_i_*.

The diameter differentiation index (*TD_i_*) represents the difference in horizontal plane between a reference tree and its neighbors. In order to reflect the difference of the DBH between the reference tree and the neighbors, *TD_i_* was set as either positive or negative [42]. When the DBH of the reference tree was larger than that of the neighbors, *TD_i_* was set positive. In this case, the reference tree has competitive advantage over its neighbors horizontally. When the DBH of the reference tree was smaller than that of the neighbors, *TD_i_* was set negative. In this case, the reference tree was less competitive compared to its neighbors. Meanwhile, the absolute values of *TD_i_* were divided into four grades: (i) small difference (0.0–0.3, i.e., *TD_i_* = 0.15); (ii) moderate difference (0.3–0.5, i.e., *TD_i_* = 0.4); (iii) large difference (0.5–0.7, i.e., *TD_i_* = 0.6); (iv) very large difference (0.7–1.0, i.e., *TD_i_* = 0.85) [30]. The absolute value of *TD_i_* was calculated as:(3)$$|{TD}_{i}|=\frac{1}{n}\sum_{j=1}^{n}\left(1-r_{ij}\right)$$
*r_ij_* is smaller DBH/larger DBH; *n* is the number of neighboring trees.

The height differentiation index (*HD_i_*) describes the variation in the vertical plane within the forest. In order to reflect the difference of the tree height between the reference tree and its neighbors, *HD_i_* was set as either positive or negative [42]. When the tree height of the reference tree was larger than that of the neighbors, *HD_i_* was set positive. In this case, the reference tree had competitive advantage over its neighbors vertically. When the tree height of the reference tree was smaller than that of the neighbors, *HD_i_* was set negative. In this case, the reference tree was less competitive compared to its neighbors in competition vertically. Moreover, the absolute values of *HD_i_* were divided into four grades: (i) small difference (0.0–0.3, i.e., *HD_i_* = 0.15); (ii) moderate difference (0.3–0.5, i.e., *HD_i_* = 0.4); (iii) large difference (0.5–0.7, i.e., *HD_i_* = 0.6); (iv) very large difference (0.7–1.0, i.e., *HD_i_* = 0.85) [30]. The absolute values of *HD_i_* were calculated as:(4)$$|{HD}_{i}|=\frac{1}{n}\sum_{j=1}^{n}\left(1-r_{ij}\right)$$
*r_ij_* is smaller height/larger height; *n* is the number of neighboring trees.

### 2.4. Data Analyses

Basic data processing was carried out with MS-Excel 2019. The statistical analysis was conducted using SPSS 20 for Windows [45]. For the variables including *M_i_*, *D_i_*, *TD_i_*, *HD_i_* because the data conformed to normal distribution and homogeneity variances, we applied one-way ANOVAs with post hoc Tukey’s test [46] to determine the difference in paper mulberry’s structural indices between every two plots (i.e., plot A vs. plot B, plot A vs. plot C, and plot B vs. plot C). Furthermore, we divided all reference trees of paper mulberry in the three sites into two categories (i.e., females and males). We then applied Student’s *t*-test for two independent samples to determine the differences in *D_i_* of female reference trees and male ones. Namely, we compared the average distances between the female reference trees and their neighbors including the female, male and other trees, respectively, and the average distances between the male reference trees and their neighbors including the male, female and other trees, respectively. Furthermore, the frequency distribution of structural indices across the three sites was graphed using Origin 8.6 [47].

## 3. Results

### 3.1. Number, Density, and Sex Ratio of Paper Mulberry

There were 41 reference trees in plot A, including 19 female and 22 male trees. The tree density was 2744 ha^−1^. There were 41 reference trees in plot B, including 21 females and 20 males. The tree density was 2622 ha^−1^. There were 43 reference trees in plot C, including 18 female and 25 male trees. The tree density was 2805 ha^−1^.

A total of 535 paper mulberry individuals (DBH ≥ 4.0 cm) were recorded, including 265 female trees, 270 male trees. In plot A, there were 180 individuals of paper mulberry (i.e., 90 females and 90 males). In plot B, there were 162 individuals of paper mulberry (i.e., 87 females and 75 males). In plot C, there were 193 individuals of paper mulberry (i.e., 88 females and 105 males). Among the three plots, plot A and C had slightly more paper mulberry males than females, but plot B had slightly more females. Overall, the sex ratio was close to 1, and there were a few more males than females.

### 3.2. Diameter and Height Distributions of Paper Mulberry and Other Tree Species

The DBH class distribution pattern of the three plots was similar for different tree species (Figure 3). In the three plots, the paper mulberry and other tree species were most distributed at the size of 4–8 cm, and the number of individuals (paper mulberry and other tree species) decreased with the DBH. The height class distribution pattern was similar between plot A and plot C, which differed from plot B in the pattern (Figure 3). The tree height distribution of paper mulberry was mainly of 8–12 m in plots A and C, but it concentrated at 4–8 m in plot B. The tree height distribution was mainly concentrated at 4–8 m for the other tree species in the three plots.

### 3.3. Structural Indices of the Paper Mulberry Communities

#### 3.3.1. The Mingling Index and the Tree–Tree Interval Index

In plot A, the relative frequency at *M_i_* = 0.00 was the highest (51.22%; Figure 4a), followed by *M_i_* = 0.25 (39.02%), and the sum of these two reached 90.24%. In plot B, the relative frequency at *M_i_* = 0.25 was the highest (46.34%), followed by *M_i_* = 0.00 (37.59%), and the sum of these two reached 80.93%. In plot C, the relative frequency at *M_i_* = 0.00 was the highest (58.14%), followed by *M_i_* = 0.25 (34.88%), and the sum of these two reached 93.02%.

Across the three sites, the relative frequency of *M_i_* = 0.00 and *M_i_* = 0.25 was the largest, which means that the neighboring trees of the reference tree were mainly paper mulberry, and there were fewer other trees (i.e., *Melia azedarach*, *Maclura tricuspidata*, *Morus alba*, *Celtis sinensis*, *Pistacia chinensis*).

We divided all of the tree–tree interval index (*D_i_*) into four categories from 0 to 4 at a step interval of 1 m (Figure 4b). The average distances between reference tree and neighboring trees were 1.62 m, 1.59 m and 1.60 m in plot A, plot B, and plot C, respectively. The average distance between the reference tree and neighboring trees in the three plots was 1.60 m.

#### 3.3.2. The Diameter Differentiation Index and the Height Differentiation Index

In plot A, the relative distribution frequency of the positive differentiation index was 73.17%, and it was higher at *TD_i_* = 0.15 and 0.40 (43.90% and 17.07%, respectively). In plot B, the relative distribution frequency of the positive differentiation index was 53.67%, and it was higher at *TD_i_* = 0.15 and 0.40 (36.59% and 12.20%, respectively). In plot C, the relative distribution frequency of the positive differentiation index was 60.47%, and it was higher at *TD_i_* = 0.15 and 0.40 (37.21% and 20.93%, respectively). There were few trees at *TD_i_* = 0.60 (2.44%, 4.88%, 2.33%, respectively) and at *TD_i_* = 0.85 in the three plots. Overall, the average diameter of paper mulberry was slightly larger than those of the neighboring trees in the three plots (Figure 5a). 

In plot A, the relative distribution frequency of the positive differentiation index was 87.80%, and it was the highest at *HD_i_* = 0.15 (73.17%). In plot B, the relative distribution frequency of the positive differentiation index was 70.73%, and it was the highest at *HD_i_* = 0.15 (70.73%). In plot C, the relative distribution frequency of the positive differentiation index was 92.69%, and it was the highest at *HD_i_* = 0.15 (82.93%). There were no trees either at *HD_i_* = 0.60 or *HD_i_* = 0.85 in the three plots, and in plot B there were no trees at *HD_i_*= 0.40 as well. Overall, the average height of paper mulberry was slightly greater than those of the neighboring trees in the three plots (Figure 5b). 

In addition, the results of ANOVAs showed that there was no significant difference between every two plots in the four structural indices (for *M_i_*: *df* = 2, *p* = 1.00 > 0.05; for *D_i_*: *df* = 2, *p* = 1.00 > 0.05; for *TD_i_*: *df* = 2, *p* = 0.98 > 0.05; for *HD_i_*: *df* = 2, *p* = 0.93 > 0.05). 

### 3.4. Spatial Distributions of Male and Female Paper Mulberry

There were 58 female reference trees and 67 male ones in the three plots. The neighboring trees around female reference trees were divided into female paper mulberry, male paper mulberry and other tree species. 

In total, there were 164 females, 26 males of paper mulberry, and 42 individuals of other tree species around female reference trees. The average distance between female reference trees and neighboring female paper mulberry, female reference trees and neighboring male paper mulberry, female reference trees and other tree species were 1.56 (±0.74) m, 1.55 (±0.70) m and 1.47 (±0.68) m, respectively. There were 186 males, 47 females of paper mulberry, and 35 individuals of other trees species around male reference trees. The average distance between male reference trees and neighboring male paper mulberry, male reference trees and neighboring female paper mulberry, male reference trees and other tree species were 1.68 (±0.91) m, 1.69 (±0.98) m and 1.59 (±0.70) m, respectively (Figure 6). 

For the female reference trees in the three plots, the average distance between them and their surrounding female paper mulberry was smaller than that for the male reference trees (Figure 6). Moreover, the distance between female reference trees and their neighbors was shorter than that of male reference trees with their neighbors in the three plots. However, no significant difference was found for the average distance between female reference trees and male reference trees (*df* = 4, *p* = 0.78 > 0.05).

Consistently, the female trees of paper mulberry were mainly surrounded by female ones, rather than male ones or the other tree species in the three sites. Likewise, male trees were mainly surrounded by male ones, rather than female ones or the other tree species in the three sites.

## 4. Discussion

### 4.1. Competition and Structural Indices of Paper Mulberry

Structural indices are often used to reflect the structural diversity of trees in a forest community [32]. Also, structural indices are used to reflect the spatial configuration, growth status and competition of reference trees in forest plots [29,43]. Our results show that the value of the mingling index was mainly concentrated at *M_i_* = 0.00 and *M_i_* = 0.25 in each plot (Figure 4a). This indicates that most of the neighboring trees of these reference trees are paper mulberry. And for paper mulberry, a smaller *M_i_* value indicates stronger competitiveness (Table 1). Thus, paper mulberry has a stronger competitiveness relative to other tree species in the paper mulberry community. Moreover, the average distance between paper mulberry and four neighboring trees in the three sample plots was mostly between 1 and 2 m (Figure 4b). This indicates a strong competition between paper mulberry and its neighbors. In horizontal and vertical planes, the sum of the frequency distribution with positive values was greater than that of negative values for both *TD_i_* and *HD_i_*, both of which concentrated at 0.15 in each plot (Figure 5). In addition, the average distances of female reference trees with their neighbors, and of male reference trees with their neighbors were all between 1 m and 2 m in each plot. Nevertheless, the mean distances (*D_i_*) between the female reference trees and their surrounding females, males and other non-paper-mulberry trees were smaller than those between the male reference tree and their three counterparts. And, the DBH of the female paper mulberry in the three plots was greater than that of males. Therefore, females appear to be more competitive than males. Collectively, our data show that paper mulberry has a competitive advantage over its neighbor trees in the native distribution range. 

Our results are consistent with other studies addressing the competitiveness of paper mulberry in the introduced areas. The previous studies have demonstrated that the invasion of paper mulberry can reduce species diversity of the local vegetation, and the cover of native tree species [16,48]. For example, paper mulberry was introduced for paper production in Ghana in 1969 and has become an invasive species since 1994 [14]. Paper mulberry invasion is thought to be a major threat to native species regeneration, and has markedly reduced the abundance of indigenous broadleaved tree species in Ghana’s forest ecosystems. Therefore, it is possible that the competitiveness in the native distribution range allows a competitive advantage over other species in invaded regions. 

The competitive advantage of paper mulberry in the study site can be primarily attributed to three factors. Firstly, paper mulberry has relatively high irradiance- and CO_2_-saturated photosynthetic rates, the maximum fluorescence, maximum quantum yield, photochemical quenching, and relative electron transport rate [49,50]. Likewise, this species presents a large variation in leaf morphology, with leaves being deeply lobed, three-five lobed, undivided, or serrate (Figure 1b). These features allow paper mulberry to have a strong adaptability to light environment. Secondly, compared with the confamilial *Morus alba*, paper mulberry shows faster decomposition of leaf litter, and hence has greater turnover of nutrients [9]. This may be conducive to the growth of paper mulberry. Thirdly, paper mulberry is a fast-growing tree species. For example, its DBH is able to reach 17.35 cm within 11 years in light of stem analysis [51]. Collectively, high photosynthetic physiological index, rapid litter turnover rate and fast tree growth enable paper mulberry to gain competitive advantage over neighboring trees in subtropical forests. 

Such high competitiveness of paper mulberry might be responsible for its wide distribution in its native range and its dominance in tree layer of forest communities. We further suggest that such a competitive advantage may facilitate the invasion of paper mulberry in many introduced regions. Nevertheless, our results do not support the competitive release hypothesis, which states that alien invasive species may be released from competition in introduced areas or habitats without competitors or with novel competitors [52,53].

### 4.2. Spatial Configuration of Female and Male Paper Mulberry

Most female paper mulberry plants, playing the role of reference trees, are primarily surrounded by female paper mulberry trees, and most male paper mulberry plants are mostly surrounded by male trees of the same species in the three plots. This indicates that spatial segregation of males and females may be common in the three paper mulberry populations. According to our investigation, for paper mulberry there are two types of population regeneration [54]. One is sexual reproduction leading to seed production. The other is clonal propagation, e.g., sprouting, which is even more common for the regeneration of paper mulberry in forest stands. The clonality may play an important role in sexual spatial segregation of paper mulberry. 

The spatial configuration of female and male paper mulberry is likely to have important ecological implications for geographical distribution. On the spatial scale, field investigations show that there are more paper mulberry plants than other species in forest stands. Both females and males present an aggregation distribution pattern. As a wind-pollinated species, the pollen grains of paper mulberry can be transferred from a patch of male trees of paper mulberry to another patch of females. Accordingly, aggregation distribution likely increases the efficiency of wind pollination, enhancing fruit-setting rates, and ultimately improving the adaptability of paper mulberry. Moreover, as a dichogamy plant [13], spatial segregation of the sexes may be an effective mechanism to encourage outcrossing and minimize inbreeding of paper mulberry population.

On the time scale, paper mulberry begins to bloom when it starts to leaf in the early spring. Most plants of paper mulberry in the stands are mainly surrounded by paper mulberry. The reference trees and the majority of their neighbors are paper mulberry with the same sex (Figure 6). Moreover, for paper mulberry, there appears a synchronization between reference trees and their neighboring trees in leafing and flowering phenology. The synchronization may be conducive to pollination because young and small leaves of paper mulberry do not block the wind-driven spread of its pollen grains.

## 5. Conclusions

There is an intense interspecific competition between paper mulberry and its neighbor tree species in subtropical deciduous broad-leaved forests, eastern China. This study is one of the first to demonstrate that paper mulberry has a competitive advantage over its neighboring trees in China in light of structural indices. Moreover, the females appear to be more competitive than the males of paper mulberry. We argue that the competitiveness of paper mulberry is an important factor responsible for wide distribution in its native range, and that it may also play a significant role in the ecological invasion of paper mulberry. In addition, because most paper mulberry plants are extensively distributed in the tropics and subtropics and rarely distributed in temperate regions in China, it is necessary to carry out a similar analysis in temperate areas in the future.

## Figures and Tables

**Figure 1 biology-12-01410-f001:**
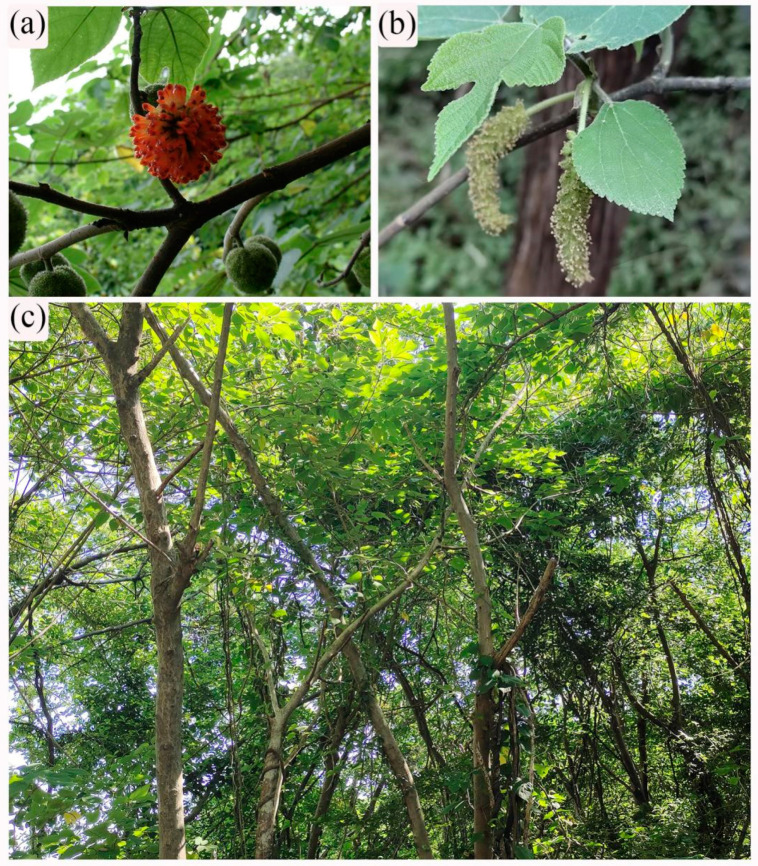
Female inflorescence of paper mulberry (*Broussonetia papyrifera*) (**a**) and male inflorescence of *B. papyrifera* (**b**) in a subtropical forest community with *B. papyrifera* (**c**) of eastern China. The photographs were taken by Guangfu Zhang.

**Figure 2 biology-12-01410-f002:**
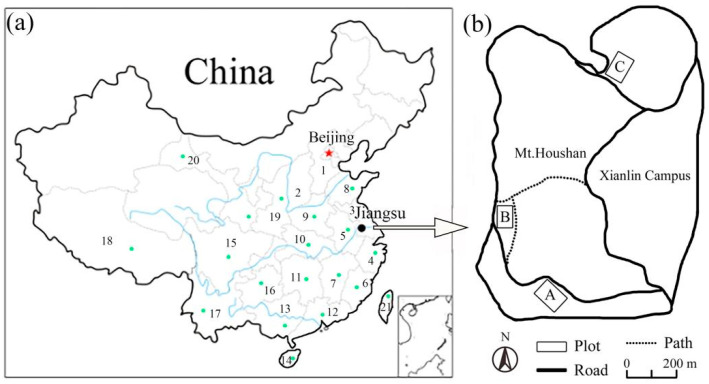
(**a**) Paper mulberry (*Broussonetia papyrifera*) occurs in 21 provinces of China, and three forest locations (**b**) in eastern China were selected for study. The provinces in the (**a**) are represented as follows: 1. Hebei; 2. Shanxi; 3. Jiangsu; 4. Zhejiang; 5. Anhui; 6. Fujian; 7. Jiangxi; 8. Shandong; 9. Henan; 10. Hubei; 11. Hunan; 12. Guangdong; 13. Guangxi; 14. Hainan; 15. Sichuan; 16. Guizhou; 17. Yunnan; 18. Xizang, 19. Shaanxi; 20. Gansu; 21 Taiwan. A, B, and C indicate plot A, plot B and plot C respectively from three deciduous broad-leaved forests with paper mulberry in Xianlinhoushan Mountain, Nanjing, Jiangsu Province, eastern China.

**Figure 3 biology-12-01410-f003:**
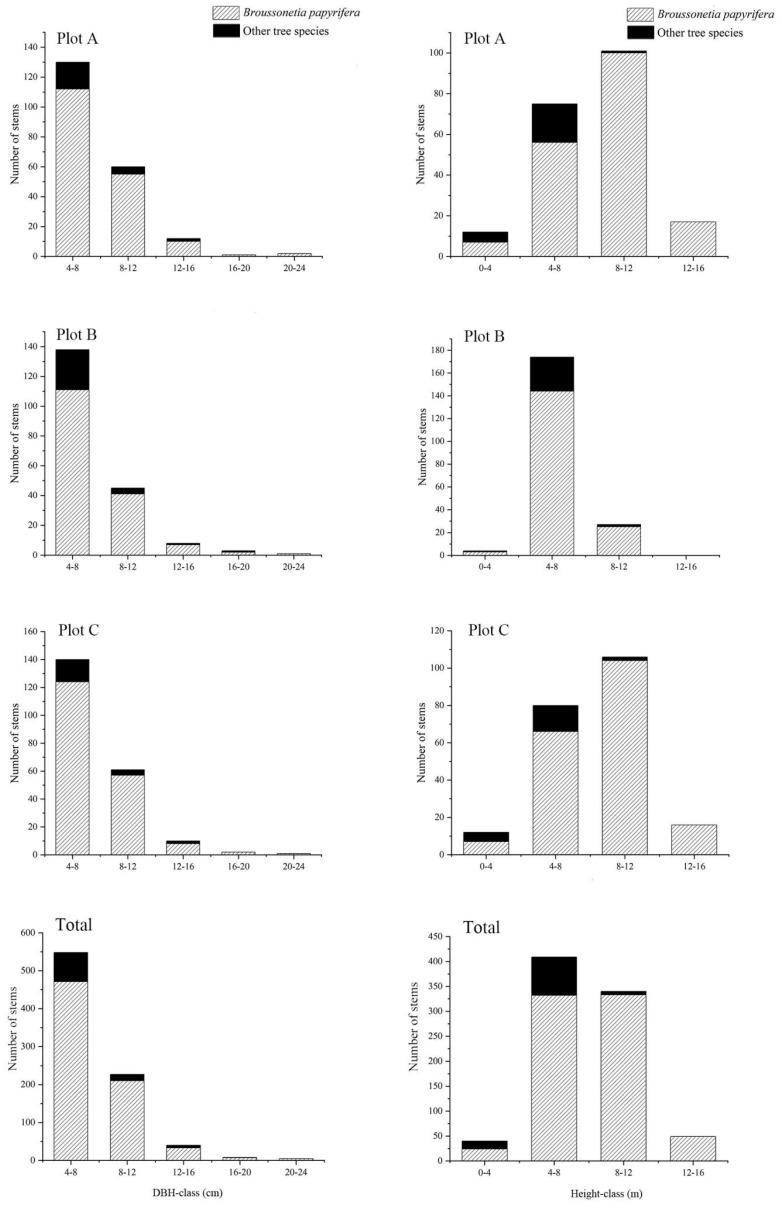
Diameter and height distributions for paper mulberry (*Broussonetia papyrifera*) and other tree species in the three plots from Xianlinhoushan Mountain in the suburban areas of Nanjing, Jiangsu Province, eastern China.

**Figure 4 biology-12-01410-f004:**
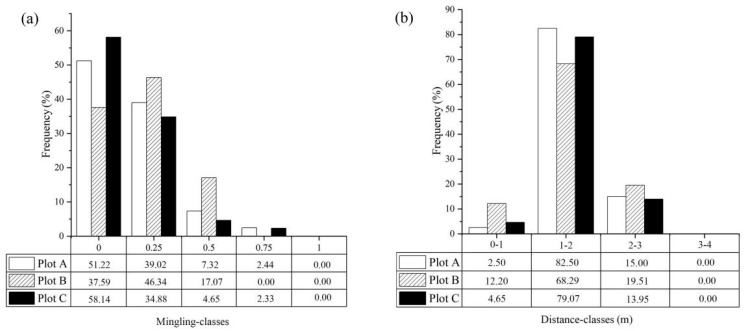
(**a**) The mingling index (*M_i_*) and (**b**) the distance to neighbors (*D_i_*) for paper mulberry (*Broussonetia papyrifera*) reference trees in the three plots from Xianlinhoushan Mountain in the suburban areas of Nanjing, Jiangsu Province, eastern China.

**Figure 5 biology-12-01410-f005:**
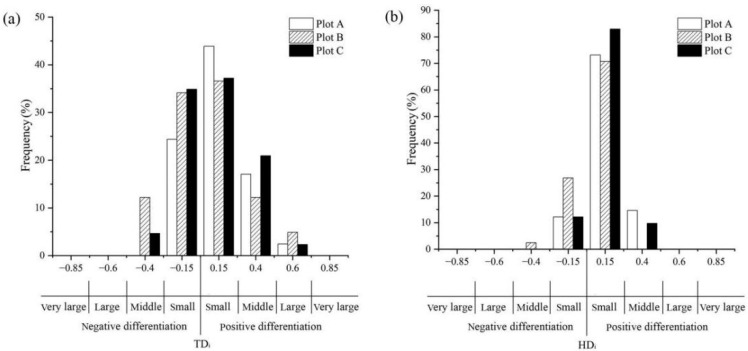
(**a**) The diameter differentiation index (*TD_i_*) and (**b**) the height differentiation index (*HD_i_*) for paper mulberry (*Broussonetia papyrifera*) in the three plots (A, B, and C) from Xianlinhoushan Mountain in the suburban areas of Nanjing, Jiangsu Province, eastern China.

**Figure 6 biology-12-01410-f006:**
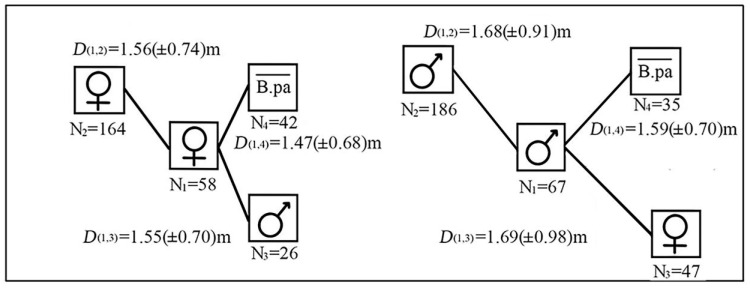
Average distance (*D*) between female/male reference trees of paper mulberry (*Broussonetia papyrifera*) and other tree species (i.e., non-paper mulberry trees) in the three plots on the Xianlinhoushan Mountain in the suburban areas of Nanjing, Jiangsu Province, eastern China. $\bar{B.pa}$ indicates non-paper mulberry trees. *D* represents the distance, and the numbers of 1 and 3 in the subscript refer to the corresponding box labeled by N_1_ and N_3_ in the nearby graph, respectively. Namely, *D*_(1,3)_ means the distance between females of paper mulberry (N_1_ = 58) and males (N_3_ = 26). The same below.

**Table 1 biology-12-01410-t001:** Summary of structural indices used to characterize *B. papyrifera* stands. This table was adapted from Sefidi et al. [44].

Structural Indices	Measures	Forest Structural Characteristics	Implication for Competition
Mixing index (*M_i_*)	Species interspersion	A value of 0 indicates a pure forest stand. A value near 1 indicates that species are highly interspersed.	For a reference tree, a smaller value indicates stronger competitiveness, and vice versa.
Tree–tree interval index (*D_i_*)	Stand density	Higher values indicate a lower stand density.	For a reference tree, a smaller value indicates stronger competitiveness, and vice versa.
Diameter differentiation index (*TD_i_*)	Tree diameter variation	A value of 0 indicates uniformity in tree diameters. A value near 1 indicates a high variation in tree diameters.	Positive value indicates a reference tree has advantage over its neighbors in competition horizontally. Higher value indicates greater competitiveness, and vice versa.
Height differentiation index (*HD_i_*)	Tree height variation	A value of 0 indicates uniformity in tree height. A value near 1 indicates high variation in tree height.	Positive value indicates a reference tree has advantage over its neighbors in competition vertically. Higher value indicates greater competitiveness, and vice versa.

## Data Availability

Data are contained within the article and Appendix A.

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
