# Peer review of "Competitive Advantage of Broussonetia papyrifera Growing in a Native Area as Suggested by Structural Diversity"

_biology, 2023, doi:10.3390/biology12111410_

Round 1
Reviewer 1 Report
Comments and Suggestions for Authors
I have not comments. Entire work is very clear for me.
However, the paper should be very well read in order to correct some typing error. For example, at r. 79 a word is repeted.
Reviewer 2 Report
Comments and Suggestions for Authors
The study entitled "Reasons for widespread distribution: structural diversity of the dioecious
Broussonetia papyrifera populations in eastern China" written by Yanrong Zhou and Guangfu
Zhang addresses B. papyrifera stands on a local scale in its native distribution range. The chosen tree species is interesting from the perspective that this species has an invasive potential. Studying invasive species ecology and community structure is important to improve models predicting the spread of the species in the future and to improve management strategies in the introduced range where this species may threaten ecosystem integrity.
However, in my opinion, the study title promises more than the study design can provide. The sampling was conducted only in the three plots on a relatively small area with consideration only on the tree layer and the four closest neighbours to the reference (centre) tree. Further, selected indices describing the structure of the B. papyrifera stands were calculated. Based on the values of the indices, the authors made conclusions on species' competitive ability. I think such conclusions are too strong as true evidence for the competitive strength of B. papyrifera is missing in this study.
Besides the study design, ideas and explanations for some phenomena throughout the manuscript are repeated. Introduction should be revised and methods description excluded. Discussion is mainly repetition of the results but in-depth explanations of the results are missing. Conclusions are too strong and without sufficient evidence.
After language editing and content corrections, this study could be interesting enough to be published in regional/national or local journals. Unfortunately, I think that the content of the study is not sufficient to be published in an international journal.
Yours sincerely,
Reviewer
Comments on the Quality of English LanguageShould be substantially improved.
Reviewer 3 Report
Comments and Suggestions for Authors
The title is suggestive, but I would like to suggest a more concise version in the future. The abstract is fine, but it does not illustrate very well the purpose of the paper.
The introduction can be improved, and Figure 1 does not have a bibliographic source, so if it is your own source, it should be specified and moved to material and method.
The results are clearly presented and the conclusions are supported by the results. I recommend a very short linguistic check.
Comments on the Quality of English Language
I recommend a very short linguistic check.
Reviewer 4 Report
Comments and Suggestions for Authors
I would like to congratulate the Authors on the great job done! Text in manuscript is balanced and was pleasure to read it. This is meaningful study and acceptable for publication after a MINOR REVISION. Make sure that the cited references within the text should be numbered.

Comments on the Quality of English LanguageSeveral corrections were added in the reviewed manuscript.
Round 2
Reviewer 2 Report
Comments and Suggestions for Authors
See the attachment.

Comments on the Quality of English LanguageLanguage should be edited.
